# Molecular Mechanisms of *Shigella* Pathogenesis; Recent Advances

**DOI:** 10.3390/ijms24032448

**Published:** 2023-01-26

**Authors:** Babak Pakbin, Wolfram Manuel Brück, Thomas B. Brück

**Affiliations:** 1Werner Siemens Chair of Synthetic Biotechnology, Department of Chemistry, Technical University of Munich (TUM), Lichtenberg Str. 4, 85748 Garching bei München, Germany; 2Institute for Life Technologies, University of Applied Sciences Western Switzerland Valais-Wallis, 1950 Sion, Switzerland

**Keywords:** pathogenesis, virulence factor encoding genes, *Shigella dysenteriae* type 1, non-Dysenteriae *Shigella* species

## Abstract

*Shigella* species are the main cause of bacillary diarrhoea or shigellosis in humans. These organisms are the inhabitants of the human intestinal tract; however, they are one of the main concerns in public health in both developed and developing countries. In this study, we reviewed and summarised the previous studies and recent advances in molecular mechanisms of pathogenesis of *Shigella* Dysenteriae and non-Dysenteriae species. Regarding the molecular mechanisms of pathogenesis and the presence of virulence factor encoding genes in *Shigella* strains, species of this bacteria are categorised into Dysenteriae and non-Dysenteriae clinical groups. *Shigella* species uses attachment, invasion, intracellular motility, toxin secretion and host cell interruption mechanisms, causing mild diarrhoea, haemorrhagic colitis and haemolytic uremic syndrome diseases in humans through the expression of effector delivery systems, protein effectors, toxins, host cell immune system evasion and iron uptake genes. The investigation of these genes and molecular mechanisms can help us to develop and design new methods to detect and differentiate these organisms in food and clinical samples and determine appropriate strategies to prevent and treat the intestinal and extraintestinal infections caused by these enteric pathogens.

## 1. Introduction

*Shigella* are facultative anaerobe, non-spore-forming, rod-shaped, non-motile and Gram-negative bacteria belonging to the *Enterobacteriaceae* family [1]. Genotypically and phenotypically, these bacteria are closely related to *Escherichia coli* strains; however, *Shigella* strains are less active in carbohydrate utilisation, by which they can be differentiated and identified biochemically from the *Escherichia coli* strains. *Shigella* strains are among the oldest human (Homo sapiens) specific pathogens, which evolved 170,000 years ago [2]. In 1888, Chantemesse and Widel isolated a bacterium from faecal samples collected from human patients with acute dysentery. Ten years later, Kiyoshi Shiga isolated and identified *Shigella* strains as humans’ main causative agents of bacillary dysentery [3]. Then, twenty years later, different serological groups of *Shigella* strains were distinguished and differentiated [1]. Using novel genetic methods, more than 46 serotypes of this bacteria have been identified and classified. Four species subgroups of *Shigella* have been identified, including *S. dysenteriae* (subgroup A; 15 serotypes), *S. flexneri* (subgroup B; 6 serotypes), *S. boydii* (subgroup C; 20 serotypes) and *S. sonnei* (subgroup D; 1 serotype) [4]. These four species are differentiated based on differences in biochemical reactions and their O antigen lipopolysaccharide. Four species of *Shigella* are widely transmitted among humans and foods via the faecal–oral route [5].

*Shigella* strains have been classified as waterborne and foodborne pathogens [6]. The natural reservoirs and hosts for these bacteria are humans and primates. *Shigella* strains are found in the faeces of infected people (or primate animals) and it can be transmitted via a vehicle such as contaminated food and water to cause any infection and diseases in humans [7]. Several outbreaks have occurred via the consumption of contaminated food and water sources. *Shigella* foodborne outbreaks are usually common when consuming foods subjected to processing by hand, exposed to a limited thermal treatment or raw foods [8]. *Shigella* strains have commonly been isolated from contaminated ground beef, oysters, potato salads, bean dip, raw vegetables and fish [9]. Person-to-person transmission of *Shigella* strains also can occur through the faecal–oral route in foodborne and waterborne outbreaks [10]. Clinical symptoms of gastrointestinal infections caused by *Shigella* species vary from mild diarrhoea for a few days by *S. sonnei* to dysentery (bloody stools with small and painful mucoid), vomiting and nausea by *S. dysenteriae*. The infective dose of *Shigella* species ranges from 10 cells of *S. dysenteriae* to 500 cells of *S. sonnei* in humans [11,12].

Shigellosis, a global health problem in developed and developing countries, is a contagious infectious disease caused by different species of *Shigella* via the consumption of contaminated food and water [13]. However, the incidence of shigellosis is also significantly high among homosexual males via person-to-person transmission [14]. It is caused by invasion of the colon, ileum and rectum epithelial cells. *Shigella* infections occur in people of all ages; however, it has been reported more in sensitive populations. More than one million deaths annually are caused by diarrhoeal diseases worldwide [15]. However, approximately 164,000 deaths are attributed to shigellosis annually, particularly in children under 5 years old. It is also estimated that more than 125 million diarrhoeal episodes are annually caused by *Shigella* species [11,13]. In addition to public health concerns generated by this disease, severe economic losses are caused by shigellosis worldwide. Investigation of the pathogenicity mechanisms of *Shigella* species helps us to determine more successful, preventive, specific and effective strategies against shigellosis [16,17]. This article will review and provide new insights into the novel aspects of molecular mechanisms of *Shigella* pathogenesis for intestinal and extraintestinal diseases in humans. We summarised these mechanisms and listed all virulence factor encoding genes associated with different pathogenesis mechanisms of Dysenteriae and non-Dysenteriae species of *Shigella*.

## 2. Bacterial Pathogenesis

A wide spectrum of microbial strains, including several bacteria, fungi, parasites and viruses, have been identified, classified and characterised as pathogens causing different types of chronic and acute infectious diseases in humans. Bacterial strains are the main causative agents of mortality and morbidity among the infected persons and hospitalised patients worldwide [18,19]. According to recent documented reports, more than 90% of infections in hospitalised patients have been caused by bacterial agents; consequently, it is estimated that a considerable number of bacterial infections occur in the general population [20]. Most microbial pathogens in humans have food product origin and are considered foodborne pathogens. These organisms contribute to foodborne diseases in humans [21]. According to the World Health Organization, foodborne illnesses are toxic or infectious diseases caused by consuming contaminated food or water [22]. Foodborne bacterial pathogens cause human intestinal and extraintestinal diseases through different pathogenicity mechanisms [23]. 

### 2.1. Bacterial Pathogenicity Definition

Bacterial pathogenicity is defined as the ability of bacterial strains to induce infection, release toxins, produce virulence and cause diseases and mortality in the host (humans). All bacterial pathogens have pathogenicity activity [24]. Specific mechanisms mediate this ability between bacterial pathogens and human host cells. Pathogenesis of a bacterial strain is defined as the mechanisms and series of events by which the bacterial cells induce and develop a morbid state or disease in the host (humans) [25]. The degree of virulence of a bacterial strain is the measure of the pathogenicity of a bacterium. Susceptibility to a bacterial infection depends on the bacteria’s virulence degree and the host’s physiologic conditions (humans) [26]. There is a complex binary interaction between the bacterial pathogen and the host throughout the development of the disease [27]. Consequently, pathogenicity mechanisms refer to the mechanisms by which the disease is developed in the host and to the pathogenesis mechanisms mediated by the bacterial strains interacting with the host. Considering the host-pathogen interactions, various virulence factors in pathogenic bacterial cells mediate the pathogenesis mechanisms in interaction with the host cells [28,29].

### 2.2. Virulence Factors in Bacterial Pathogenesis

Understanding the virulence factors is important in studying bacterial pathogenesis in establishing disease in the host [30]. As previously discussed, diseases caused by Enterobacteriaceae family members are mediated via intestinal and extraintestinal pathogenesis mechanisms. These mechanisms are regulated and developed via several mechanism-specific virulence factors of the pathogenic bacteria [31,32]. Virulence factors in bacterial pathogens are divided into 6 main classes including (1) membrane proteins, (2) capsule, (3) secretory proteins, (4) outer membrane proteins, (5) biofilm formation and (6) iron acquisition. The membrane proteins class includes adhesion, invasion, colonisation, and surface virulence factors. Secretory proteins are subclassified into immune response inhibitors, toxins, and transport groups [33,34,35].

Regarding the pathogenesis mechanism of each bacterial pathogen, the expression and release of these virulence factors has been regulated and induced by the bacterial strain [20]. Adhesion factors lead to binding to the host epithelial cells and mucosal layers. Different types of adhesin proteins are expressed and released on bacterial cell surfaces. Adhesion can be mediated by either fimbrial or afimbrial structures [36,37,38]. Protein secretion pathways also play a key role in the pathogenesis of Gram-negative bacterial pathogens, and there are seven distinct types of secretion systems in these bacteria. Host cell invasion, intracellular multiplication and cell-to-cell spread are prominent virulence factors in the pathogenicity mechanisms of invasive bacterial pathogens such as *Shigella* species [16,17]. Several pathogenic bacteria release some specific polypeptides with toxic effects on host cell processes and structures. These toxins allow the pathogens to cause damage and grow within the host cells [39]. Toxins are secreted via lysis of the bacterial cell (endotoxin) or through a specific secretion system (exotoxin). Bacterial pathogens can also evade the host immune system by employing specific mechanisms [40,41]. Regarding the mechanism of pathogenesis, the type of virulence factors and the symptoms of the disease, *Shigella* species are categorised into two distinct pathogenic groups: Dysenteriae and non-Dysenteriae species [16,42]. In the following sections of this paper, we will investigate the pathogenic molecular mechanisms and virulence factors of these two pathogenic groups separately.

## 3. Pathogenic Molecular Mechanisms of Non-Dysenteriae *Shigella* Species Pathogenesis

### 3.1. Non-Dysenteriae Shigella Species

The epidemiological studies of shigellosis showed that the species of *S. sonnei* and *S. flexneri* are the main causes of this disease [43]. *S. dysenteriae* type 1 causes more severe and fatal disease in human as an extraintestinal pathogen, which has been investigated in the next section. Regarding the epidemiological studies, *S. sonnei* and *S. flexneri* are more prevalent for endemic shigellosis in high-income and low-income (developed and developing) countries, respectively [11,12]. All species of *Shigella* except *S. dysenteriae* type 1 have been considered as non-Dysenteriae species of *Shigella*, causing intestinal shigellosis [16]. Regarding public health concerns, three major characteristics have been identified for shigellosis: first, as a paediatric disease with more than 60% of the cases between the ages of 1 to 5 years old; second, as a general diarrhoeal disease with more than 150 million cases each year; and third, the deadly disease, mostly in young children and infants [11,12,13]. Non-Dysenteriae *Shigella* species cause the first and second groups. Non-Dysenteriae species of *Shigella* are invasive bacteria and cause mild diarrhoea in humans via intracellular multiplication and gastrointestinal epithelial cell manipulation mechanisms [6,16,17].

### 3.2. Molecular Mechanisms of Non-Dysenteriae Shigella Species Pathogenesis 

*S. flexneri* and *S. sonnei* (non-Shiga-toxin producing strains) are mainly responsible for non-Dysenteriae disease (mild diarrhoea) caused by *Shigella* strains [5,8,16]. The pathogenesis of *Shigella* possesses unique and multiple well-established pathogenic mechanisms mediated by seven distinct steps, including attachment to the epithelial host cell, entry to the cell, host cell autophagy evasion, vacuole formation, vacuole rapture, intracellular life and immune response [44,45]. *Shigella* employs these seven steps, and releases different effector proteins to damage, invade and suppress the immune system of the host gastrointestinal epithelial cells, contributing, eventually, to mild diarrhoeal symptoms in the patient [3,46]. Infection by *Shigella* initiates in the colon by reaching the underlying submucosa through transcytosis and passing through microfold cells. As the first barrier, Mucin glycol-protein is remodelled and glycosylated by *Shigella* during the attachment process [43,47,48]. Gel-forming mucin is induced by proinflammatory cytokines such as tumour necrosis factor-α (TNFα) and interferon-gamma (IFN-γ). Mu5AC factor, secreted by *Shigella* strains, stimulates specific gel-forming mucin and induces the accumulation of gel-like structures on the host cell surface, leading to bacterial invasion. This mechanism is dependent on the presence of the type III secretion system (T3SS) in the bacterial cell [3,12,46]. 

The virulence and pathogenesis of *Shigella* strains require T3SS, the most important pathogenicity mechanism of *Shigella* species. T3SS is a needle-like molecular structure in the bacterial cell wall encoded on the Mxi-spa locus, located on a 220 kb virulence plasmid (pWR100) in *Shigella* strains [49]. All virulence factor encoded genes providing different mechanisms of Dysenteriae and non-Dysenteriae *Shigella* species are described in Table 1. It provides a direct channel between the host cytoplasm and the bacterial cell to inject several bacterial protein effectors through this syringe-like structure. Mxi-spa locus encodes *Mxi* (A-N) and spa genes, expressing the components of T3SS structure [50,51]. *Shigella* strains use T3SS, mainly, to induce its uptake into the host epithelial cells, escape from the vacuole, mediate cell-to-cell spreading and manipulate the host cell processes. These mechanisms are mediated through different protein effectors [46,49]. After the initial attachment by accumulating a gel-like structure on the surface of the host cells, complementary attachment is mediated by a specific outer membrane protein (*IcsA*) encoded by *VirG* gene. The binding site of the *IcsA* protein on the host cell is Neural Wiskott-Aldrich syndrome protein (N-WASP), a WASP family and CDC-42 dependent compound mediating the actin nucleation in the host epithelial cell cytoplasm through the Arp2/3 complex [25,44,52]. *IpaB* and *IpaC* are involved in pore formation in host cell. *IpaD*, as a hydrophilic protein, binds with T3SS tips and blocks the needle pore. Following the sensing of cholesterol and sphingomyelin on the host cell membrane, *IpaD* stimulates *IpaB* in the T3SS tip to mediate the invasion of the host cell [44,51]. Key protein effectors such as *VirA, IPgD, IpaA, IpaB* and *IpaC* secreted through T3SS into the host cell mediate epithelial cell signalling, cellular uptake, cytoskeletal rearrangements and lysis of the endocytic vacuole [45,53,54]. *VirA*, *IPgD* and *IpaA* are involved in microtubule and actin destabilisation to develop the invasion into the phagosome and establish the endocytic vacuole [44,55]. *IpaH, IpaB, IpaC* and *IpaD* are the protein effectors responsible for phagosome escape in the host cell cytoplasm [44,52,56]. 

In addition to T3SS, type VI secretion system (T6SS) is also employed by *S. sonnei* strains and *tss* (A-K) genes encode this mechanism. The type II secretion system (T2SS), encoded by *gsp* (A-O) genes, is also used by *S. boydii* Sb227 and *S. dysenteriae* Sd197 strains [44,57,58]. 

The intracellular dissemination of *Shigella* species through host epithelial cells is mediated via the manipulation of cell membrane and cytoskeleton. Some T3SS effector proteins, such as *IpgB1* and *IpgB2*, are responsible for these mechanisms through the regulation of Rho family GTPases in host cells [44,45]. These effectors are guanine nucleotide exchange factors and are defined and grouped together by the WxxxE motif (tryptophan-xxx-glutamic acid). These effectors mimic the low molecular weight GTPases and activate the signalling cascades in the host cells. They are mainly encoded in T3SS gene domains [50,52]. However, WxxxE sequences have also been detected in TLR domains of both *Shigella* species and human host cells. WxxxE family proteins such as *IpgB2* interreact with human RhoA via inducing conformational changes. The activation of the Rho protein family induces host cell contraction via cell membrane ruffling through the activation of GTP-binding proteins such as Rac1 and Cdc42 [44,46,51]. It has also been shown that WxxxE effectors can bind to the host engulfment and modulate the host immune system. *IpgB1* is also involved in the subversion of actin cytoskeleton dynamics. *IpgB2* activates the nuclear factors through the activation of Rho-associated kinases. *IpgB1* and *IpgB2* also have a key role in the tight junctions by the pathways of unidentified signalling because of their Rho and Rac mimicking abilities [44,45,50]. After the cell entry, pyroptosis is mediated through interleukins and activated by nod- and Toll-like receptors contributing to the activation of inflammatory caspases. Pyroptosis causes membrane rupture in the host cells, contributing to inflammatory response and ion venting. However, *Shigella* bacterial cells can prevent host cell death and mediate intracellular life by *IpaH* family effectors such as *IpaH9.8*, *IpaH7.8*, *IpaH1.4* and *IpaH4.5* [44,46]. It should be noted that different proteins in host cells have key roles in *Shigella* pathogenesis, including N-WASP, ARP2/3, NLR, TLR, Integrin, Talin, Actin, Rho family proteins, TNF-α, IFN-γ, Mu5AC, NF-κB and interleukins [44].

After entering into the host epithelial cells, *Shigella* strains promote their survival by using different protein effectors and subverting the host cell processes. *IcsA* protein (*VirG*) triggers actin nucleation and induces the actin microfilament growth at the one pole of the organism, mediating the movement of the bacterial cell through the host cell cytoplasm [59,60]. The *VirA* effector also destabilises the microtubules and enables the bacteria to control the actin polymerization leading to efficient spread of the bacterial cells throughout the host cell. This mechanism provides the intracellular motility and dissemination characteristics of *Shigella* bacterial cells [45,47]. *Shigella* strains employ three main mechanisms to prevent host cell death and provide the replication niche for bacterial infection. *IpaB* effector has been shown to induce host cell cycle arrest via targeting Mitotic Arrest Deficient 2 Like 2 (MAD2L2, an anaphase inhibitor) protein to prevent the host cell turnover. Preventing the host cell detachment, *OspE* protein effector interacts with the integrin-linked kinase. *IpgD* also prevents apoptosis in the host epithelial cell via the activation of Akt proteins and stimulation of phosphoinositide-3-kinase [44,45,47]. *Shigella* bacterial cells must evade the innate immune responses to persist inside the host cells, and *Osp* genes mediate this mechanism. *OspF* induces the dephosphorylation of the mitogen-activated protein kinases required for the regulation of nuclear factor kappa B (NF-κB) gene expression in the host cell. *OspG* and *OspI* effectors also bind the E2 protein and inhibit the activation of NF-κB. *OspF* and *OspB* effectors reduce the interleukin-8 (IL-8) secretion level in the host cells. *OspD2* inhibits necrosis, and *OspC3* mediates the inhibition of pyroptosis in the infected host cells [44,61,62]. As a non-*Osp* effector, *ipaH* can interact with the splicing factors and inhibits the expression of inflammatory cytokines. The *IpaH* effectors family also are involved in other host cellular processes, including protein degradation, cell cycle and endocytosis [63,64]. *Shigella* lipopolysaccharide (LPS), composed of core polysaccharides, O-antigen and lipid A and encoded by *gtr* genes, also mediates intracellular spread and resistance to host cell defence [65,66]. The expression of iron uptake genes is critical for intracellular growth, and this protective mechanism is mediated by aerobactin proteins encoded by *iuc* (A-D) genes [44,52,67].

Autophagy is a protective and defence mechanism of the host cells which enables them to survive during infections via the degradation of macromolecules to recycle the damaged organelles and the energy in the cell cytoplasm [68]. This defence mechanism is mediated by autophagy related proteins (ARP). The *IcsB* effector can be recognised as ARP, masks the region that is unmasked by ARP and inhibits the autophagy mechanism of the host cell [69,70]. Non-Dysenteriae *Shigella* species inside the host cells secrete different toxins and toxic compounds. Two main types of exotoxins, including *Shigella* enterotoxin 1 (ShET1) and *Shigella* enterotoxin 2 (ShET2), are secreted by non-Dysenteriae *Shigella* species. ShET1 is chromosomally encoded by *set1A* and *set1B* genes. This exotoxin is an iron-dependent toxin (55 KDa), mainly released by *S. flexneri* strains, and predominantly accounts for the early phase of diarrhoea in shigellosis. ShET2 is a plasmid-borne toxin (inv plasmid) encoded by the *senB* gene and also responsible for the early phase of diarrhoea [12,71,72]. Other toxins have also been described for the *Shigella* species, including protein involved in intestinal colonisation and *Shigella* IgA-like protease homology toxins encoded by *pic* and *sigA* genes, respectively. Pic is a protease non-cytotoxic toxin, secreted via type V secretion system (T5SS), mainly released by *S. flexneri* strains and belonging to the serine protease autotransporters of the Enterobacteriaceae family (SPATEs). SigA is a protease cytotoxin, also secreted through T5SS and belonging to the SPATEs. This toxin is a serine protease involved in the accumulation of intestinal fluid during shigellosis [43,44,73]. 

Since *Shigella* species are intracellular pathogens, there are several host proteins and engulfment that affect the pathogenesis mechanisms of these bacterial pathogens. The invasion of host epithelial cells by *Shigella* species requires a comprehensive adaptation of the bacterial pathogen to the new environment through accessing different metabolic cofactors and nutrients mediating intracellular growth. *Shigella* also employs some specific structures in host cells, such as N-WASP, Arp2/3 complex and actin molecules during their pathogenesis mechanisms [58,59]. Several in vivo studies revealed that human host cells elaborate specific mechanisms, such as immune response signalling and autophagy against *Shigella* invasion, intracellular growth and dissemination through the host cells. In the initial steps of the epithelial host cell infection by *Shigella*, host cells release IL-8 and pro-inflammatory cytokines, including IL-1β and IL-18 by macrophages, induce inflammation and activate the immune cells, including neutrophils and natural killer cells, against *Shigella* invasion [50,53,54]. Host cells mediate the autophagy mechanism against pathogenic bacterial intracellular dissemination. Autophagy in human epithelial cells is mediated through recruiting TNF receptor-associated factor 6, ubiquitin, LC3, NDP52, ATG5, P62, calpains, caspase-8 and Beclin-1 factors during *Shigella* infection and intoxication (by Shiga-toxin) [44,46,49,50,54]. *Shigella* species elaborate specific mechanisms to suppress these antimicrobial strategies throughout intracellular growth, dissemination and activities which are also comprehensively described in this review paper.

## 4. Pathogenic Molecular Mechanisms of Dysenteriae *Shigella* Species Pathogenesis

### 4.1. Dysenteriae Shigella Species Diseases 

*S. dysenteriae* is one of the four serological groups of *Shigella* bacteria. Type 1 of *S. dysenteriae* is the only bacterial strain of *Shigella* species considered as the clinical group of Dysenteriae *Shigella* species and differentiated from other species due to the production of Shiga toxins (Stxs); however, some studies recently reported the production of Stxs by *S. sonnei* isolated from clinical samples [8,74]. *S. dysenteriae* type 1 is also considered one of the leading causes of bacillary dysentery or shigellosis and is transmitted to humans through the ingestion of contaminated water or food. It is also transmitted person-to-person via the faecal–oral route [44,75]. *S. dysenteriae* infection and intoxication are primarily more prevalent in lower-middle- and low-income countries, especially among children, contributing to long-term diarrhoea and malnutrition due to intestinal protein loss [76,77]. *S. dysenteriae* serotype 1 is also known for producing explosive pandemics with high fatality rates, including pandemics which have occurred in the past fifty years in central America, south Asia, and central and east areas of Africa [12,13,78]. Another public health concern regarding *S. dysenteriae* type 1 is that this organism is resistant to all the currently used antibiotics [5]. The incubation period of shigellosis caused by *Shigella* non-dysenteriae species ranges from 1 to 4 days, and typically for *S. dysenteriae* type 1 it is up to 8 days. In the first phase, *S. dysenteriae* type 1 causes watery diarrhoea, mediated by the secretion of an enterotoxin, for 1 or 2 days. Then, the organism invades the large intestine epithelial cells and produces cramps, fever, tenesmus and bloody diarrhoea. At this phase, *S. dysenteriae* type 1 is isolated from the stool culture of more than 50% of the cases [79,80,81]. As a Shiga-toxin-producing strain, *S. dysenteriae* type 1 mainly causes haemolytic uraemic syndrome (HUS) in patients. Shiga toxin secreted by *S. dysenteriae* type 1 accounts for more than 2,800,000 acute diseases, leading to approximately 3900 HUS cases annually. Around 2–7% of *S. dysenteriae* infections cause HUS in humans [82].

### 4.2. Molecular Mechanisms of Dysenteriae Shigella Species Pathogenesis 

*S. dysenteriae* is a normal microflora of the human gastrointestinal system; however, it is considered one of the most important public health concerns, and a major cause of bloody diarrhoea and HUS (*S. dysenteriae* type 1) in humans [74]. The serotype 1 of this organism is closely related to *Escherichia coli* serotype O157: H7, regarding the fact that both of these organisms can secret Stxs [83,84]. Foodborne outbreaks caused by *S. dysenteriae* type 1 are highly associated with the consumption of undercooked processed foods and vegetable salads [74,85]. HUS is the only disease caused by *S. dysenteriae* type 1 infection, and the typical symptoms of this disease are renal failure, thrombocytopenia and microangiopathic haemolytic anaemia [58]. The fatality rate of this disease is 35%. *S. dysenteriae* type 1 adheres to the human colonic mucin and attaches to the epithelial cells by using the same mechanisms and virulence factor genes employed by other species [12,13]. After attachment and adherence, Stxs are secreted by this organism [74]. In addition to the presence of Stxs encoding genes, there are some minor differences between the virulence factors of *S. dysenteriae* type 1 and other species of *Shigella* bacteria. *S. dysenteriae* type 1 uses type 2 secretion system (T2SS) as an effector delivery system to inject protein effectors into the host cell. This needle-like structure is encoded and expressed by *gsp* (A-O) genes in *S. dysenteriae* type 1 [86,87]. *S. dysenteriae* type 1 also uses a specific heme uptake system as a protective virulence factor encoded by *Shu* genes. This mechanism mediates the direct binding of heme-containing proteins and the secretion of hemophores to provide iron for the organism during the pathogenesis and survival processes. This mechanism also protects the DNA of *S. dysenteriae* type 1 from heme-mediated oxidative damage [44,67]. 

Stxs, also known as Shiga-like toxins, verocytotoxins and verotoxins, are lethal toxins first isolated from *S. dysenteriae* in 1975 [88,89]. In addition to *S. dysenteriae* type 1, some other bacterial pathogens secret these toxins, such as some *E. coli* serogroups (and more than 200 serotypes), including O157, O26, O91, O103, O104, O145 and *S. sonnei*, which has recently been reported [90,91]. Two types of Stxs have been identified, including Stx1 and Stx2. Additionally, different variants of these toxins have been characterised, such as Stx1 (c-d) and Stx2 (c-k). Stx2 is 56% identical to Stx1, considering the amino acid sequence. Stx members are AB5 proteins composed of the A subunit (32 KDa), which is noncovalently bound to five B subunits (each one is 7.7 KDa) [90,92,93]. A and B subunits of Stxs are encoded by chromosomal *stxA* and *stxB* genes. The pentameric structure of Stx (B5) binds to globotriaosylceramide (GB3), its receptor on the surface of the host Intestinal and kidney epithelial cells. After binding, Stx enters the host cell through the endocytosis mechanism and is trafficked into the Golgi apparatus, where the A subunit is activated [94,95,96,97]. A subunit is the active component of the Stx and is composed of two disulfide-bonded fragments (A1-A2). Subunit A presents proteolytic activity, deactivates 28 s rRNA and inhibits peptide synthesis in host cells, leading to apoptosis, necrosis and cell death [11,12,89,98,99]. Stxs are responsible for HUS, haemorrhagic colitis and bloody diarrhoea in the human host, since they target both intestinal and kidney epithelial cells [13,44,74]. All pathogenesis mechanisms of Dysenteriae and non-Dysenteriae *Shigella* species are summarised in Figure 1.

## 5. Limitations and Perspectives

*Shigella* is one of the most important bacterial infectious challenges in food safety and public health, causing diarrhoea and extra-intestinal disorders including hemolytic colitis and haemorrhagic uremic syndrome in humans. So far, several mechanisms of pathogenesis and virulence factors of *Shigella* species have been defined and specified. However, various types of new protein effectors and the main roles of different effectors have not been known and identified clearly [44,51,58]. Bacterial Quorum sensing systems (QS) have recently been known as novel virulence factors specified for *Enterobacteriaceae* family pathogens such as *Salmonella*, *Escherichia* and *Shigella* species. QS is a bacterial cell-to-cell communication mechanism contributing to the adaptation of the organism to the new environment. Through this pathogenesis mechanism, bacterial strains release specific QS molecules as signals which are received and detected by other strains and allow them to respond to challenging environmental conditions [100]. These mechanisms have also been characterised for *Shigella* species, especially *S. flexneri*; however, more studies need to be implemented to deeply understand the key role of QS molecules and mechanisms in *Shigella* pathogenesis [101]. On the other hand, using QS inhibitors has widely been developed as a practical, affordable and efficient anti-bacterial strategy in food industries and clinical treatments [102]. 

## 6. Conclusions

*Shigella* species are one of the main concerns in public health, and they cause mild diarrhoea, hemolytic colitis and haemorrhagic uremic syndrome in humans. Regarding the severity and type of the caused diseases and the presence of different virulence factor encoding genes, species of *Shigella* are categorised into two groups: Dysenteriae and non-Dysenteriae species. The pathogenesis of non-Dysenteriae *Shigella* species includes attachment, invasion, vacuole escape, intracellular motility, toxin secretion and host cell interruption. The pathogenesis mechanisms of non-Dysenteriae and Dysenteriae species of *Shigella* are mediated by the expression of effector delivery systems (T3SS and T6SS), different protein effectors, toxins, iron uptake genes and the secretion of the Shiga-toxin (only for *S. dysenteriae* type 1). *S. dysenteriae* type 1 uses T2SS as the effector delivery system. Shiga-toxins contribute to HUS and bloody diarrhoea in humans, with a high rate of fatality. Regarding the specific genes in each group, invasion and Shiga-toxin encoding genes can be considered as the target genes to detect and differentiate Dysenteriae and non-Dysenteriae species of *Shigella* in food and clinical samples. Additionally, considering the different pathogenicity molecular mechanisms and virulence factor encoding genes in each clinical group, distinct and relative therapeutic strategies can be used to prevent and treat the intestinal and extraintestinal infections caused by these bacterial enteric pathogens.

## Figures and Tables

**Figure 1 ijms-24-02448-f001:**
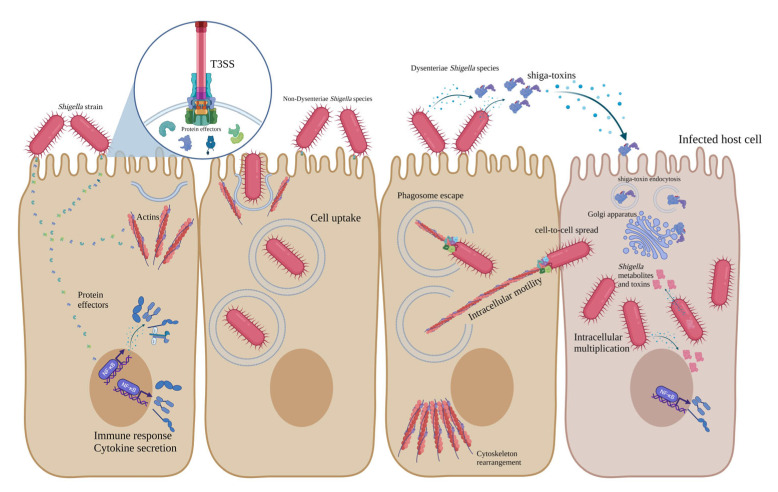
Schematic representation of pathogenesis mechanisms of Dysenteriae and non-Dysenteriae species of *Shigella*.

**Table 1 ijms-24-02448-t001:** Virulence factor encoding genes in Dysenteriae and non-Dysenteriae *Shigella* species.

Class	Virulence Factor Gene	Function	Type (D/ND) ^1^
Effector delivery	*GspC, GspD, GspE, GspF, GspG, GspH, GspI, GspJ, GspK, GspL, GspM, GspO*	T2SS structure	D
	*MxiA, MxiC, MxiD, spa9, spa13, spa15, spa24, spa29, spa33, spa40, spa47*	T3SS structure	D/ND
	*TssA, TssB, TssC, TssD, TssE, TssF, TssG, TssH, TssI, TssJ, TssK, TssL,* *TssM*	T6SS structure	ND
Effectors	*IcsA*	intracellular motility	D/ND
	*IcsB*	autophagy evasion and actin destabilisation	D/ND
	*IpaA*	actin depolymerisation	D/ND
	*IpaB*	endocytosis, phagosome escape, cell apoptosis inhibition	D/ND
	*IpaC*	adhesion to host cell and phagosome escape	D/ND
	*IpaH*	inflammatory response suppression, phagosome escape, protein degradation, cell cycle intervention and pyroptosis induction	D/ND
	WxxxE effectors *(IpgB1 and IpgB2)*	cell entry, invasion, host cell process manipulation, inflammation regulation, actin organisation, vacuole formation, apoptosis and tight junction disruption	D/ND
	*IpgD*	actin destabilisation, endocytic vacuole lysis and apoptosis prevention	D/ND
	*ospB*	immune response suppression	D/ND
	*ospC*	cell pyroptosis inhibition	D/ND
	*ospD*	cell pyroptosis and necrosis inhibition	D/ND
	*ospE*	bacterial dissemination	D/ND
	*ospF*	immune response suppression	D/ND
	*ospG*	immune response suppression	D/ND
	*ospI*	immune response suppression	D/ND
	*virA*	actin destabilisation and endocytic vacuole lysis	D/ND
Toxins	*Set1A*	encoding *Shigella* enterotoxin 1, early diarrhoea	ND
	*Set1B*	encoding *Shigella* enterotoxin 1, early diarrhoea	ND
	*senB*	encoding *Shigella* enterotoxin 2, early diarrhoea	ND
	*gtrA*	encoding lipopolysaccharide, intracellular spread, immune modulation, host cell defence inhibition	D/ND
	*gtrB*	encoding lipopolysaccharide, intracellular spread, immune modulation, host cell defence inhibition	D/ND
	*pic*	encoding protein involved in intestinal colonisation toxin, SPATEs, haemagglutinin activity	ND
	*sigA*	encoding *Shigella* IgA-like protease homology, SPATEs, intestinal fluid accumulation	ND
	*stxA*	encoding Shiga-toxin subunit A, proteolytic activity, peptide synthesis inhibition, haemorrhagic colitis and the haemolytic uremic syndrome	D
	*stxB*	encoding Shiga-toxin B subunits, binding to the cellular receptors, haemorrhagic colitis and the haemolytic uremic syndrome	D
Metabolic	*iucA, iucB, iucC, iucD, iutA*	encoding aerobactin, iron uptake	D/ND
	*ShuA, ShuS, ShuT, ShuU, ShuV, ShuY, ShuX*	encoding hemophores, heme uptake system, DNA protection against oxidative damages	D

^1^ D, Dysenteriae *Shigella* species; ND, non-Dysenteriae *Shigella* species.

## Data Availability

We confirm that all data supporting the findings of this study are available within the article.

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
