# Peer review of "Molecular Mechanisms of Shigella Pathogenesis; Recent Advances"

_ijms, 2023, doi:10.3390/ijms24032448_

Round 1

Reviewer 1 Report

In this review the authors discussed Shigella pathogenesis. The review includes the following points: a) Bacterial pathogenesis  b) molecular mechanisms of non-Dysenteriae Shigella species pathogenesis.  c)molecular mechanisms of Dysenteriae Shigella species pathogenesis

Since the review consider the recent advances in this field, I found that the recent data is not included

Major points

1) The authors should discuss about Shigella entry, and the use of TSS3 as IpgB1 in the entry and invasion

2) Role of the WxxxE motif present vacterial effectors  in Shigella  pathogenesis. Is the same motif present in non-pathogenic Shigella?

3) Role of host protein engulfment in Shigella pathogenesis? 

4) Model systems used in studying Shigella pathogenesis.

Author Response

Dear Reviewer 1,

Thank you very much for your precise reviewing and your practical and helpful comments. All revisions have been addressed completely according to your comments as described below:

- The discussion regarding the Shigella entry, using T3SS, IpgB1 and IpgB2 genes in entry and invasion by Shigella species are added into the manuscript.

- Role of WxxxE motif in pathogenesis mechanisms of Shigella species also added into the manuscript.

- Roles of host proteins during the pathogenesis of Shigella have been highlighted and more explained in details in the manuscript.

- We summarized these mechanisms and listed all virulence factor encoding genes associ-ated with different pathogenesis mechanisms of Dysenteriae and non-Dysenteriae species of Shigella. Model systems to study these mechanisms are included in the introduction and abstract sections of the study.

Reviewer 2 Report

The manuscript by Pakbin et al. entitled “Molecular mechanisms of Shigella pathogenesis; recent advances” described brief information on various virulence factors and pathogenicity mechanisms of the Dysenteriae and non-Dysenteriae Shigella spp. Overall, the manuscript requires revision to justify its publication as follows:

Comments

1.      Line 29-34, Virulence factors expression in Enterobacteriaceae of animal and plant pathogens are regulated by Quorum-sensing systems? Please add such important information, i.e., Microorganisms 10 (2022) 2211; Molecules 27 (2022) 7584. Also, add such information in section 2.

2.      Please add one illustration of the mechanism of pathogenesis and prevention strategies for Shigella spp.

3.      The author can add one brief section on potential recent strategies for treating bacterial pathogenes in animals, humans, and plants, such as using quorum sensing inhibitors, i.e., Biotechnology Advances 37 (2019) 68-90.

4.      Add one section on “ major challenges or limitations and perspectives”. 

Author Response

Dear Reviewer 2,

Thank you very much for your precise reviewing and your practical and helpful comments. All revisions have been addressed completely according to your comments as described below:

- Quorum-sensing systems as a novel virulence factors in Shigella species have been explained and added into the manuscript in a separate section.

- One illustration containing schematic representation of pathogenesis mechanisms of Dysenteriae and non-Dysenteriae spe-cies of Shigella is added into the manuscript.

- Using quorum sensing inhibitors as one of the novel efficient strategies against Shigella infection is added into the manuscript in a separate section.

- The limitations and perspectives section is added into the manuscript containing recent challenges and novelties in Shigella pathogenesis.

Reviewer 3 Report

This study is about new insights into the novel aspects of molecular mechanisms of Shigella pathogenesis for intestinal and extraintestinal diseases in humans.There are the appropriate and adequate references to related and previous work. English used is correct and readable. However, there are some comments:

Major point:

The manuscript is not well organized and comprehensively described. The sections are limited. The authors can add some more sections or divided the existence sections to more subsections.

Minor points:

The abstract can be written better with organized sections: aim, methodology, results, conclusion...

The search methodology can be added.

Some figures are needed for this manuscript.

Author Response

Dear Reviewer 3,

Thank you very much for your precise reviewing and your practical and helpful comments. All revisions have been addressed completely according to your comments as described below:

  • The manuscript is divided into more subsections and one section including limitations and perspectives is added into the manuscript.
  • The abstract is revised according to your comment.
  • The search methodology is based on reviewing recent papers and our original publications. It is mentioned in the aim of the study in introduction and abstract sections.
  • One illustration containing schematic representation of pathogenesis mechanisms of Dysenteriae and non-Dysenteriae spe-cies of Shigella is added into the manuscript.

Round 2

Reviewer 1 Report

The authors have not addressed my comments:

1) The role of WxxxE motif in Shigella pathogenesis

2) Host proteins that impact Shigella pathogenesis

Author Response

Dear reviewer,

Thank you very much for your precise review and practical comments,

The role of WxxxE motif in Shigella pathogenesis and host proteins that impact Shigella pathogenesis are comprehensively discussed in two separate paragraphs in section 3.2 of the manuscript according to your comments.

Reviewer 2 Report

Accept

Author Response

Dear reviewer,

Thank you very much.

Round 3

Reviewer 1 Report

No further comments